# Barriers and facilitators to physicians' telemedicine uptake during the beginning of the COVID-19 pandemic

Jack D. Watson[1,2], Bridget Xia[3], Mia E. Dini[4], Alexandra L. Silverman[5], Bradford S. Pierce[6], Chi-Ning Chang[7], Paul B. Perrin[3,4]*

1 Informatics, Decision Enhancement, and Analytic Sciences Center, VA Salt Lake City, Salt Lake City, Utah, United States of America, 2 Department of Internal Medicine, Division of Epidemiology, University of Utah School of Medicine, Salt Lake City, Utah, United States of America, 3 School of Data Science, University of Virginia, Charlottesville, Virginia, United States of America, 4 Department of Psychology, University of Virginia, Charlottesville, Virginia, United States of America, 5 McLean Hospital/Department of Psychiatry, Harvard Medical School, Belmont, Massachusetts, United States of America, 6 James A. Haley Veterans Affairs Medical Center, Tampa, Florida, United States of America, 7 School of Education, Virginia Commonwealth University, Richmond, Virginia, United States of America

* perrin@virginia.edu

## Abstract

Despite decades of low utilization, telemedicine adoption expanded at an unprecedented rate during the COVID-19 pandemic. This study examined quantitative and qualitative data provided by a national online sample of 228 practicing physicians (64% were women, and 75% were White) to identify facilitators and barriers to the adoption of telemedicine in the United States (U.S.) at the beginning of the COVID-19 pandemic. Logistic regressions were used to predict the most frequently endorsed (20% or more) barriers and facilitators based on participant demographics and practice characteristics. The top five reported barriers were: lack of patient access to technology (77.6%), insufficient insurance reimbursement (53.5%), diminished doctor-patient relationship (46.9%), inadequate video/audio technology (46.1%), and diminished quality of delivered care (42.1%). The top five reported facilitators were: better access to care (75.4%), increased safety (70.6%), efficient use of time (60.5%), lower cost for patients (43%), and effectiveness (28.9%). Physicians' demographic and practice setting characteristics significantly predicted their endorsement of telemedicine barriers and facilitators. Older physicians were less likely to endorse inefficient use of time ($p < 0.001$) and potential for medical errors ($p = 0.034$) as barriers to telemedicine use compared to younger physicians. Physicians working in a medical center were more likely to endorse inadequate video/audio technology ($p = 0.037$) and lack of patient access to technology ($p = 0.035$) as a barrier and more likely to endorse lower cost for patients as a facilitator ($p = 0.041$) than providers working in other settings. Male physicians were more likely to endorse inefficient use of time as a barrier ($p = 0.007$) than female physicians, and White physicians were less likely to endorse lower costs for patients as a facilitator ($p = 0.012$) than physicians of color. These findings provide important context for future implementation strategies for healthcare systems attempting to increase telemedicine utilization.

**Data availability statement:** Data are publicly available as a supplementary attachment to this manuscript.

**Funding:** The author(s) received no specific funding for this work.

**Competing interests:** The authors have declared that no competing interests exist.

## Author summary

Many physicians treated patients via telemedicine to maintain social distancing during the COVID-19 pandemic. We surveyed a group of U.S. physicians to identify barriers and facilitators of using telemedicine at the beginning of the pandemic. Lack of patient access to technology necessary for telemedicine was the top barrier reported by the physicians. Improving patient access to care was the top facilitator. We also found that the physicians' age, gender, race, and practice setting influenced their opinions. Older physicians were less likely to report wasting time (such as troubleshooting technological issues) and increased possibility of making medical errors as barriers than younger physicians. Male physicians were more likely to report wasting time (such as troubleshooting technological issues) as a barrier than female physicians. White physicians were less likely to report cheaper cost as a facilitator than minority physicians. Physicians working in a medical center were more likely to report poor quality of telemedicine technology and lack of patient access to technology necessary for telemedicine as barriers and cheaper cost for patients as a facilitator. These findings may be helpful for healthcare systems to consider when implementing telemedicine practice in the future.

## Introduction

The World Health Organization (WHO) declared a pandemic related to a new strain of the coronavirus (COVID-19) in mid-March of 2020 [1]. Government entities, businesses, and other institutions (e.g., healthcare systems) sought ways to prevent the spread of infection, keep their employees and patrons safe, and comply with new pandemic-related regulations for social distancing and lockdowns [2]. One key strategy adopted by many institutions was a rapid shift to online technologies; much of what was once done in person was now conducted virtually [3].

Healthcare, especially, saw a significant and swift change. Elective procedures and routine care were postponed while healthcare systems were predicted to be overwhelmed by COVID-19 patients; medical supplies were scarce, and front-line healthcare workers were placed at increased risk for COVID-19 exposure [4]. Several major healthcare groups including the Veterans Health Administration and the Mayo Clinic Health System promptly converted to providing a large portion of their care through telemedicine, a means of delivering healthcare services remotely using a telecommunication device often connected via internet (e.g., computer, tablet, or phone) [5–8].

During the early stages of the COVID-19 pandemic, telemedicine emergency room visits increased by 50% compared to the same period in 2019 and, by the end of March, increased by a staggering 154% [9]. However, this rapid change may have been driven more from necessity than provider preference or enthusiasm for telemedicine. Indeed, as little as 3.72% of all clinical work prior to the pandemic was delivered via telemedicine [10], and clinician attitudes for and acceptance of (or lack thereof) telemedicine was the most significant factor predicting telemedicine uptake [11].

A growing body of literature has highlighted the efficacy of telemedicine treatment and its importance amidst the changing healthcare landscape [10–12]. While many healthcare providers reported satisfaction with telemedicine and a general willingness to continue using the modality [11], some research has indicated a predicted post-pandemic decrease in telemedicine use among physicians [10]. Indeed, the mainstream use of telemedicine hinges heavily on clinician enthusiasm for the modality and removing barriers to its implementation while highlighting its numerous advantages [13]. Thus, it is important to understand what variables

during the pandemic facilitated or impeded the use of telemedicine as well as potential implications for implementation and clinical use.

The growing literature on telemedicine has demonstrated a number of factors instrumental in determining attitudes, perceptions, and uptake of telemedicine delivery. Prior telemedicine education and experience have been linked to more favorable opinions of telemedicine [11,14,15]; however, this may be due more to repeated use and familiarity with telemedicine than training [10]. Reliable and flexible technology, dedicated resources for telemedicine, appropriate training, supportive leadership attitudes, and increased access to care have all been cited as factors contributing to more positive attitudes toward telemedicine [10,12,16,17]. Conversely, higher cost, lack of familiarity with technology, inequitable access, unsupportive regulations, and reimbursement issues have all been connected with reticence toward telemedicine [10,12,16,17].

Thus, the current study examined quantitative and qualitative data provided by 228 practicing physicians to identify barriers and facilitators to the adoption of telemedicine in the U.S. at the beginning of the COVID-19 pandemic. Physicians' personal and practice characteristics were also collected to help identify which characteristics were associated with the likelihood to endorse a specific barrier or facilitator to telemedicine use.

## Methods

### Participants

Physicians of all specialties were recruited across the U.S., irrespective of specific geographic location within the U.S. Physicians were invited by email to participate in the study using directories of hospital and health clinic websites, professional organizations, social media groups, and professional newsgroups. Data were collected over 11 weeks from 12 May 2020–25 July 2020. Eligibility requirements were: (a) 18 years of age or older and (b) licensed and practicing (i.e., actively seeing patients) as a physician in the U.S. A total of 850 individuals were contacted via email to participate (with invitations also posted to online groups of physicians). Forty-six invitations returned as undeliverable. A total of 315 people (39.2% of those invited) opened the survey, with 21 leaving after viewing the information sheet. Participant data were reviewed to determine eligibility and missingness (with incomplete data or those not meeting eligibility deleted) for a total sample size of 228, licensed, practicing physicians. This response rate is acceptable for studies using similar methods of recruitment techniques [18,19]. Data for the current study were collected as part of a larger study [10]. The initial invitation was crafted to avoid indicating the study's focus on telemedicine to reduce self-selection bias. Telemedicine was defined as "the use of real-time audio (e.g., telephone) and/or video conferencing technology to provide medical services."

### Ethics statement

The study was approved by the Virginia Commonwealth University Institutional Review Board (IRB) under protocol HM20019315 to ensure it was conducted ethically and in compliance with all federal, state, and local regulations concerning research involving human participants. Because the study was designated exempt by the IRB, an information form was presented to participants rather than an informed consent document.

### Measures

Participants were asked to provide demographic (e.g., age, gender, race) and practice-related (e.g., medical specialty, location, population density) information. They were asked to respond to several questions regarding their telemedicine use and select any "advantages or incentives

for using telemedicine in your practice to treat patients" and "barriers or deterrents to telemedicine in your practice to treat patients." The list of questions regarding demographic and practice characteristics (i.e., the predictors for the current study) were based on previous literature on telemedicine and telepsychology barriers, facilitators, or use. S1 Table appendix provides the survey responses.

Within the survey, the selectable options of advantages/barriers were researcher-generated from the previous literature on telemedicine barriers, facilitators, and the likelihood of use [10–12,14–17]. Similarly, all sociodemographic or practice characteristics were selected due to previous research indicating potential relationships with telemedicine use [10–12,14–17]. The final list of barriers and facilitators, including both the initial researcher-generated selectable options and the additional categories added during the qualitative analysis, can be seen in S1 Table.

## Data analysis

All analyses were conducted in *R* [20]. Descriptive statistics were calculated with percentages indicating the proportion of physicians who endorsed the various barriers and facilitators of telemedicine use. Endorsement of barriers and facilitators was defined as survey respondents choosing each option with select-all-that-apply. The options not selected were automatically considered as not endorsed by the respondents. We then ran a series of logistic regressions to predict, using participant demographics and practice characteristics, the most frequently endorsed (20% or more) barriers and facilitators with not endorsing the barrier or facilitator as the reference category. Predictors included the continuous variable age and binary variables of gender (man vs. woman), race (White/European American race/ethnicity vs. non-White individuals), medical practice settings (medical center vs. all other settings), and geographic location (urban vs. suburban and rural). For these groupings, medical center was defined as either an academic medical center or a Department of Veterans Affairs medical center, and suburban physicians were grouped with rural physicians primarily due to the sample size in an attempt to detect unique features related to urbanicity (as opposed to rural and suburban characteristics). The reference groups for these binary variables were woman, non-White individuals, all other settings, and suburban and rural areas, respectively. Chi-squared and Nagelkerke pseudo-R squared statistics were calculated to test the overall significance of the logistic regression models with all predictor variables.

To appropriately code the qualitative "other" responses for both advantages/facilitators and barriers/deterrents, we conducted a content analysis of the participants' responses [21] that is similar to descriptive content methods used in prior qualitative research on barriers and facilitators to telepsychology use [22]. We used a blended deductive and inductive coding procedure [23,24]. The original 16 barrier categories and 15 facilitator categories were generated by researchers with extensive expertise in the field of telehealth technologies and relied on prior literature on the most frequently researched barriers and facilitators to telemedicine. This research framework was used to identify these categories, establish guidelines for the types of responses that would fall under each category, and appropriately code participant responses into the researcher generated categories. Further, an inductive coding approach was used to derive concepts and themes that extended beyond the original 31 researcher generated categories. This resulted in the addition of 2 barrier/deterrent categories and 1 advantage/facilitator category. The miscellaneous category was created for codes that did not fit into the main themes [25]. This residual category was used to house codes that were either deemed not common enough to generate a new thematic category or did not fit into the deductive categories in our coding framework. This method of categorizing codes has been used in previous studies on telehealth use [22].

The two initial raters were psychology PhD students familiar with the literature on telemedicine use and independently reviewed the 62 "Other [Please specify]" responses (20 advantages/facilitators and 42 barriers/deterrents). The two raters then reviewed the list of researcher-generated categories and created mutually agreed upon definitions for each category to assist in coding. The raters also generated two additional barrier/deterrent categories: "inadequate patient technological literacy" and "not suitable for certain patients or types of care" and one additional advantage/facilitator category: "miscellaneous." Finally, one participant gave multiple "Other [Please specify]" responses which were separated into individual units of meaning for a final count of 63 "Other [Please specify]" responses with 20 advantages/facilitators and 43 barriers/deterrents. The final list of categories with the addition of the three categories created by the raters totaled 18 barriers and 16 facilitators (S1 Table).

The raters then coded the 63 "Other [Please specify]" responses resulting in an interrater reliability of 85.4%. Of the 63 "Other [Please specify]" responses, two advantages/disadvantages were removed as "nonsensical or unable to be coded." One barrier/deterrent was also listed in the advantages/facilitators section; this response was also removed from the advantages/facilitators section. This resulted in 60 "Other [Please specify]" responses being coded into the 34 barrier and facilitator categories. A third independent rater who was also a psychology PhD student served as a tiebreaker for the items on which the first two raters did not agree, choosing only from the two categories endorsed by the first two raters. The final, tie-broken list was then reintegrated with the full list of responses obtained from the survey and analyzed. Example responses for the "Other" category are provided in S2 Table. The study's anonymized data are available in S1 Data.

## Results

The sample included 228 physicians who were an average of 46.14 years old (*SD*=10.12). A greater proportion of the physicians were women (64%), identified as White (75%), and were practicing in urban and suburban settings (92.5%). There was low racial/ethnic diversity among physicians and a low percentage of physicians working in rural areas. The sample characteristics are summarized in Table 1. The medical specialties reported by the physicians can be found in Table 2. Since the physicians were allowed to select more than one answer for the medical specialty question on the survey, the total number of responses exceed our sample size of 228.

The most frequently endorsed barriers and facilitators can be seen in S3 Table. The five most frequently endorsed barriers were: lack of patient access to technology (endorsed by 77.6% of physicians), insufficient insurance reimbursement (53.5% of physicians), diminished doctor-patient relationship (46.9%), inadequate video/audio technology (46.1%), and diminished quality of delivered care (42.1%). The five most frequently endorsed facilitators were: better access to care (75.4%), increased safety (70.6%), efficient use of time (60.5%), lower cost for patients (43%), and effectiveness (28.9%). Logistic regressions were run only on the most frequently endorsed barriers and facilitators (20% or more), and these results can be seen in S4 Table and S5 Table, with overall model statistics presented in S6 Table.

Only two of the overall models were significant, the barrier of inefficient use of time (p < 0.001) and the facilitator of lower cost for patients (p = 0.002). Within the context of all other variables included in the models, older physicians were less likely to endorse inefficient use of time (*p* <.001 for the individual odds ratio) and potential for medical errors (*p* = 0.034) as barriers to telemedicine use compared to younger physicians. Physicians working in a medical center were more likely to endorse inadequate video/audio technology (*p* = 0.037) and lack of patient access to technology (*p* = 0.035) as barriers compared to those working in other settings, and male physicians were more likely to endorse inefficient use of time (*p* = 0.007)

**Table 1.** *Sample Characteristics* (N=228).

| Characteristics | (N=228) |
|---|---|
| Age M (SD) | 46.14 (10.12) |
| Years in Practice M (SD) | 18.32 (10.00) |
| Gender N, (%) | |
| Man | 82 (36%) |
| Woman | 146 (64%) |
| Race/Ethnicity N, % | |
| White/European-American | 170 (75%) |
| Black/African-American | 6 (2.6%) |
| Hispanic/Latino | 9 (3.9%) |
| Asian/Asian-American | 31 (13.6%) |
| Multiracial/Multiethnic | 8 (3.5%) |
| Other | 3 (1.3%) |
| American Indian/Alaska Native/Native American | 1 (0.4%) |
| Geographic Setting of Practice N, % | |
| Urban | 152 (66.7%) |
| Suburban | 59 (25.9%) |
| Rural | 17 (7.5%) |
| Primary Practice Setting N, % | |
| Hospital | 58 (25.4%) |
| Veterans Affairs Medical Center | 7 (3.1%) |
| Academic Medical Center | 90 (39.5%) |
| Health Maintenance Organization | 1 (0.4%) |
| Individual Practice | 6 (2.6%) |
| Group Practice | 33 (14.5%) |
| Outpatient Treatment Facility | 7 (3.1%) |
| School/University | 8 (3.5%) |
| Other | 18 (7.9%) |
| Number of Providers in Practice N, % | |
| One | 8 (3.5%) |
| Two to Five | 47 (20.6%) |
| Six to Ten | 52 (22.8%) |
| Eleven to Twenty | 29 (12.7%) |
| Twenty-one to Fifty | 21 (9.2) |
| More than Fifty | 70 (30.7) |
| Not Reported | 1 (0.4%) |

as a barrier to telemedicine uptake than female physicians (S4 Table). White physicians were less likely to endorse lower cost to patients ($p = 0.012$) as a facilitator to telemedicine use than non-White physicians, whereas physicians working in a medical center ($p = 0.041$) were more likely to endorse lower cost to patients as a facilitator than physicians working in other settings (S5 Table).

## Discussion

This study examined quantitative and qualitative data provided by 228 practicing physicians in the U.S. to identify barriers and facilitators to the adoption of telemedicine at the beginning of the COVID-19 pandemic. The top five endorsed barriers were: (1) lack of patient

**Table 2.** *Medical Specialties* (N=258).

| Medical Specialty | (N=258) |
| --- | --- |
| Allergy and Immunology | 4 (1.8%) |
| Anesthesiology | 4 (1.8%) |
| Clinical Genetics and Genomics | 1 (0.4%) |
| Colon and Rectal Surgery | 1 (0.4%) |
| Diagnostic Radiology | 3 (1.3%) |
| Emergency Medicine | 18 (7.9%) |
| Family Medicine | 21 (9.2%) |
| General Surgery | 5 (2.2%) |
| Internal Medicine | 34 (14.9%) |
| Interventional Radiology and Diagnostic Radiology | 2 (0.9%) |
| Neurological Surgery | 3 (1.3%) |
| Neurology | 4 (1.8%) |
| Obstetrics and Gynecology | 12 (5.3%) |
| Ophthalmology | 8 (3.5%) |
| Orthopedic Surgery | 2 (0.9%) |
| Otolaryngology – Head and Neck Surgery | 9 (3.9%) |
| Pathology | 3 (1.3%) |
| Pediatrics | 56 (24.6%) |
| Physical Medicine and Rehabilitation | 6 (2.6%) |
| Psychiatry | 9 (3.9%) |
| Public Health and General Preventive Medicine | 1 (0.4%) |
| Radiation Oncology | 6 (2.6%) |
| Thoracic and Cardiac Surgery | 2 (0.9%) |
| Vascular Surgery | 1 (0.4%) |
| Other | 43 (18.9%) |

Participants could select all that apply, so the total frequency is greater than the sample size.

access to technology, (2) insufficient insurance reimbursement, (3) diminished doctor-patient relationship, (4) inadequate video/audio technology, and (5) diminished quality of delivered care. The top five endorsed facilitators were: (1) better access to care, (2) increased safety, (3) efficient use of time, (4) lower cost for patients, and (5) effectiveness. These findings are largely consistent with literature prior to the COVID-19 pandemic [11,12,14,15] and during the COVID-19 pandemic [10,16,17,26] and will add to the growing body of literature on providers' perceptions of barriers and facilitators to telemedicine use. Some of the barriers (e.g., patients' technology difficulty, limited physical examination due to virtual format) identified at the beginning of the pandemic [27,28] continued to be reported throughout the pandemic [29–32]. Additionally, while there appeared to be increased concerns about the virtual format impacting provider-patient relationship [29,31,32], adequate telemedicine training and increased equipment accessibility were positively associated with telemedicine use [33]. Telemedicine has transformed the healthcare landscape in the wake of the COVID-19 pandemic [10–12]. Sustained adoption of telemedicine is warranted, given it has been shown to improve healthcare access for underserved populations [34], reduce costs [35], and improve patient outcomes [36]. The immediate uptake and wide implementation of telemedicine during the COVID-19 pandemic were in part due to the removal of legislative restrictions and reimbursement constraints, highlighting the importance of having supportive policies in place to sustain the continued use of telemedicine [37,38].

The present study identified a number of important demographic and practice-related predictors of physicians' perceptions of telemedicine use. The Nagelkerke pseudo-$R^2$ statistics in the logistic regression models had a wide range, suggesting that the predictors explained 0.4%-14.8% of the variance in barrier endorsement and between 1.4%-11.0% of the variance in facilitator endorsement. As a result, these model effect sizes were largely in the small- (≤0.01) to medium- (≤0.09) sized range. Only two of the overall models including all relevant study predictors (age, gender, race, medical practice location, and urbanicity) were significant. The relatively low explanatory power of the models may be due to (a) a lack of critical predictors in the model, (b) low statistical power due to the smaller sample size, or (c) the inclusion of predictors that do not add significant explanatory power to the model. Thus, despite the relative lack of significance for the majority of the models, we examined the strength of individual predictors.

Within these models, older physicians reported fewer barriers to telemedicine than younger physicians, particularly inefficient use of time and potential for medical errors. It is possible that career stage might be the primary force behind these differences; however, no research to date has explored why this discrepancy might exist. Older physicians may have a more established caseload, regular patients, and greater experience navigating issues that arise due to treatment or telemedicine. Seniority may also come with additional benefits (e.g., better work schedule, supervisory roles) that may make the use of telemedicine more convenient.

Physicians working in a medical center were more likely to endorse inadequate video/audio technology as a barrier and more likely to endorse lower cost for patients as a facilitator than providers working in other settings. It is possible that physicians in medical centers may have trouble finding dedicated workspaces or tools for telemedicine appointments [16] or may be frustrated by a lack of resources during telemedicine appointments that would be available during a typical in-person appointment (e.g., imaging); however, no research to date has investigated this possibility. The likelihood to endorse lower cost to patients could be related to several factors including ancillary fees that are avoided when using telemedicine (parking, transportation, etc.) or possibly a reduced fee structure for telemedicine appointments.

Male physicians were more likely to endorse inefficient use of time as a barrier than female physicians. There is some research suggesting female providers also take longer on average with their patients than male providers [39], suggesting more willingness to have a longer patient encounter. The additional time needed to start a telemedicine appointment may be especially salient for male providers. Additional research on gender-based differences in telemedicine preference is severely lacking leaving us to hypothesize other possible explanations. Male physicians might generally prefer in-person visits or grow impatient more quickly with troubleshooting difficulties, resulting in a perception that telemedicine wastes time. Patient distractibility (e.g., providing care for a child while in a telemedicine appointment) may also be influencing the difference between male and female providers. It is possible that the way in which the telemedicine program is implemented may add additional burden to the process (e.g., a telemedicine clinic that enables providers from a major metropolitan medical center to treat patients who are still required to go in-person to a rural telemedicine medical center as opposed to a direct telemedicine to home program).

Finally, White physicians were less likely to endorse lower costs for patients as a facilitator. While the specific reason for this difference is unknown (and little research has been conducted that might provide meaningful insight), it is possible that White physicians might serve a more affluent clientele. Particularly if the provider is on retainer or the patient pays out of pocket, there might be little reason to see a significant difference in cost. This could also be a function of service area or practice setting with non-White physicians being more likely to work in community-based clinics or on a sliding scale.

This study was conducted during the pandemic public health emergency when there was a rapid uptake in telemedicine use; the state of telemedicine has changed much since then. While Medicare has permanently expanded telemedicine coverage for behavioral telehealth services, many telemedicine flexibilities were temporary and have expired [40]. Currently, telehealth regulations for private insurers vary state by state [41], and many private insurers have changed reimbursement policies around telemedicine despite adopting greater flexibility during the pandemic. These changes in reimbursement underline the importance of continued advocacy work to inform policy and legislative changes to improve the sustainability of telemedicine in a post-pandemic context.

Interestingly, in the present study, diminished quality of delivered care (38.60%) was one of the most endorsed barriers, and higher quality of delivered care (5.30%) was the least endorsed facilitator, suggesting that physicians were concerned about the quality of care delivered via telemedicine at the beginning of the pandemic. Their perception contrasts with telemedicine research on the lack of impact of digital care delivery on patient-provider alliance [42,43], patient perceptions of quality of care [44], physician perceptions of increasing satisfaction with telemedicine and quality of care [45], and higher continuity of care associated with telemedicine use especially for chronic disease management [46]. Thus, physician attitudes toward telemedicine may have shifted post-pandemic given the positive healthcare outcomes illustrated by digital care delivery during the pandemic. Additionally, offering tailored training programs and implementing appropriate telehealth technologies at work may mitigate resistance from physicians who are skeptical about the effectiveness of digital care delivery.

Digital literacy has been shown to be a social determinant of health affecting healthcare outcomes [47,48]. Greater digital literacy was associated with increased telehealth use during the pandemic [49]. Individuals from disadvantaged groups are more likely to experience reduced access to telemedicine as a result of decreased digital literacy [50–52]. Targeted programs addressing digital health literacy skills may reduce disparities in telemedicine use by helping patients build necessary digital skills and improving equitable access to telemedicine [50]. To facilitate digital inclusion and bridge the digital divide, it is important to fund initiatives focusing on improving broadband infrastructure and increasing affordability of internet access [53].

## Limitations and future directions

While the current study contributed several novel findings to the burgeoning telemedicine literature, it has some limitations. Since data were collected at the beginning of the COVID-19 pandemic, it is important to contextualize the findings and recognize the transformative role telemedicine played during this critical transition period for the healthcare field. As a result, generalizability of the findings is limited, as certain perceived barriers or facilitators identified by physicians at the beginning of the pandemic may be less relevant in the post-COVID era (e.g., virtual visits facilitated infection prevention and control). Thus, all results must be interpreted through the lens of the pandemic and its many challenges, and future research should be conducted to examine how telemedicine, its implementation, and demand for it may have changed.

Several groups of people were underrepresented in the current study. The sample was heavily White- (75%) and female- (64%) biased. Further, only 17 of the 228 physicians were practicing in a rural location, an area that often faces profound and unique challenges for telemedicine (e.g., internet access, poverty, lack of providers). The lack of response from rural providers made the grouping of practice location (urban vs suburban and rural) difficult. While the current study did not uncover a significant predictive relation between urbanicity and the various barriers and facilitators, it is possible that, had the rural sample been larger,

we might have uncovered significant results for rurality (rural vs urban/suburban). Though the current study's sampling process had a response rate comparable to many similar studies, certain groups of physicians may have been more likely to complete the study, such as those with more flexible time to do so, those whose email addresses were a part of common medical listservs, and those who were scientifically minded and open to participating in research. Future studies should include a larger more diverse sample with multiple sampling approaches (e.g., in person, physical mail, advertisements, etc.) to aid in generalizability.

It is quite possible the use of telemedicine may face unique difficulties with certain patient populations, treatment modalities, or therapeutic techniques (and this was indeed noted in several of the "Other [Please specify]" responses). Recent studies have shown that the uptake of telemedicine varies by specialty [54] and clinical conditions [55]. While the current study did collect data on provider specialty, it was difficult to classify specialties meaningfully for the statistical analyses due to the diversity of specialties represented by the sample. Future studies examining barriers and facilitators to telemedicine uptake should consider provider specialty to assist in targeted programs to increase telemedicine use where it may be more difficult to implement but still beneficial for patients.

Another limitation of the study is that participants' free text responses were integrated into the existing themes, which impacted the richness and nuances of the qualitative findings. For example, while inadequate technological literacy was not listed in the survey as a barrier for physicians to choose from, it does not mean that this barrier was not perceived as important by the physicians; if it were presented in the survey, some physicians might have selected it. Future research may consider conducting qualitative interviews with physicians to examine their perceptions and opinions of telemedicine use with a focus on the impacts of the digital divide on access to telemedicine.

This study returned a number of non-significant predictors of barriers and facilitators to telemedicine uptake. There could be manifold reasons for the lack of significant findings including a true lack of effect, poor statistical power, item or scale constructs may have inherent measurement error due to imprecise wording, or there may be additional unknown methodological confounds obscuring real results. Future research may wish to investigate these predictors in a larger sample with greater power. It is possible that with a larger or more diverse sample, results might be different, with some additional categories being significantly endorsed, although the effect sizes likely would have been small-sized and possibly not as robust or important as those found in the current study. This study focused only on physicians; however, patient care is frequently multidisciplinary, multimodal, and team-based. Thus, future research, particularly for group practices, hospitals, medical centers, or other healthcare institutions with large teams of providers, should include a more diverse group that is representative of the multidimensionality of healthcare (e.g., nurses, allied health professionals, psychologists) which might also provide the ability to gather a much larger sample.

Finally, the research on barriers and facilitators to telehealth is still in its relative infancy, especially after the COVID-19 transformation of telehealth services. This study highlighted a number of gaps in the literature regarding what might underpin provider suspicion of or affinity for telemedicine. Future research may wish to examine, for instance, why male providers perceive of telemedicine as an inefficient use of time or whether physicians believe telemedicine appointments routinely lack vital resources that might be available for in-person visits.

## Conclusion

This study examined quantitative and qualitative data provided by 228 practicing physicians in the U.S. to identify barriers and facilitators to telemedicine uptake during the beginning of

the COVID-19 pandemic. Physicians endorsed lack of patient access to telemedicine technology as the primary barrier to telemedicine use and increased access to care as the primary facilitator. The current study also identified seven other frequently endorsed barriers and seven other frequently endorsed facilitators as areas for targeted intervention to assist in the implementation of telemedicine. Additionally, the current study uncovered several important demographic and practice characteristics for predicting the likelihood to endorse a particular barrier or facilitator during the beginning of the COVID-19 pandemic. These results contribute to the growing literature on telemedicine uptake and implementation, highlighting the profound need for continued research on how telemedicine may be perceived or implemented across specialties, patient populations, and treatment modalities. Since the expansion in telemedicine use will continue to influence care provision beyond the pandemic, future research should examine perceived post-pandemic barriers and facilitators of telemedicine use and develop strategies to address personal and systemic barriers and support delivery of patient-centered care via telemedicine.

## Supporting information

**S1 Table. Final list of barriers and facilitators.**
(DOCX)

**S2 Table. Sample "Other" responses.**
(DOCX)

**S3 Table. Percentage endorsed of each telemedicine barrier and facilitator.**
(DOCX)

**S4 Table. Variables in logistic regressions for barriers.**
(DOCX)

**S5 Table. Variables in logistic regressions for facilitators.**
(DOCX)

**S6 Table. Chi-squared and Nagelkerke pseudo-R squared.**
(DOCX)

**S1 Data. Publicly available data file.**
(SAV)

## Author contributions

**Conceptualization:** Jack D. Watson, Bradford S. Pierce, Paul B. Perrin.

**Data curation:** Bradford S. Pierce, Paul B. Perrin.

**Formal analysis:** Jack D. Watson, Mia E. Dini, Alexandra L. Silverman, Chi-Ning Chang.

**Investigation:** Jack D. Watson, Bradford S. Pierce, Paul B. Perrin.

**Methodology:** Bradford S. Pierce, Paul B. Perrin.

**Project administration:** Jack D. Watson, Bradford S. Pierce, Paul B. Perrin.

**Supervision:** Paul B. Perrin.

**Validation:** Mia E. Dini, Alexandra L. Silverman, Chi-Ning Chang.

**Visualization:** Jack D. Watson.

**Writing – original draft:** Jack D. Watson, Bridget Xia.

**Writing – review & editing:** Jack D. Watson, Bridget Xia, Chi-Ning Chang, Paul B. Perrin.

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
