## [Decision Letter · Decision Letter 0]

12 Sep 2024

PDIG-D-24-00296

Barriers and facilitators to physicians’ telemedicine uptake during the beginning of the COVID-19 pandemic

PLOS Digital Health

Dear Dr. Perrin,

Thank you for submitting your manuscript to PLOS Digital Health. After careful consideration, we feel that it has merit but does not fully meet PLOS Digital Health's publication criteria as it currently stands. Therefore, we invite you to submit a revised version of the manuscript that addresses the points raised during the review process.

Please submit your revised manuscript within 60 days Nov 11 2024 11:59PM. If you will need more time than this to complete your revisions, please reply to this message or contact the journal office at digitalhealth@plos.org. Please include the following items when submitting your revised manuscript:

We look forward to receiving your revised manuscript.

Kind regards,

Calvin Or, PhD

Section Editor

PLOS Digital Health

Journal Requirements:

Additional Editor Comments (if provided):

Reviewers' comments:

Reviewer's Responses to Questions

**Comments to the Author**

1. Does this manuscript meet PLOS Digital Health’s publication criteria? Is the manuscript technically sound, and do the data support the conclusions? The manuscript must describe methodologically and ethically rigorous research with conclusions that are appropriately drawn based on the data presented.

Reviewer #1: Yes

Reviewer #2: Yes

Reviewer #3: Yes

2. Has the statistical analysis been performed appropriately and rigorously?

Reviewer #1: Yes

Reviewer #2: No

Reviewer #3: Yes

3. Have the authors made all data underlying the findings in their manuscript fully available (please refer to the Data Availability Statement at the start of the manuscript PDF file)?

Reviewer #1: No

Reviewer #2: Yes

Reviewer #3: No

4. Is the manuscript presented in an intelligible fashion and written in standard English?

PLOS Digital Health does not copyedit accepted manuscripts, so the language in submitted articles must be clear, correct, and unambiguous. Any typographical or grammatical errors should be corrected at revision, so please note any specific errors here.

Reviewer #1: Yes

Reviewer #2: Yes

Reviewer #3: Yes

5. Review Comments to the Author

Please use the space provided to explain your answers to the questions above. You may also include additional comments for the author, including concerns about dual publication, research ethics, or publication ethics. (Please upload your review as an attachment if it exceeds 20,000 characters)

Reviewer #1: This original research manuscript entitled “Barriers and facilitators to physicians’ telemedicine uptake during the beginning of the COVID-19 pandemic” is a well written manuscript assessing providers attitudes towards telepsychology. Overall, the findings from the study are important and quite relevant in the current healthcare climate. I generally think this manuscript is a good one, though there are a few questions that arise when reading it. 

There are several points the investigative team should consider:

1/ Who were the physicians enrolled in the study? It is not entirely clear. There is a section on Table 1 that would discusses the primary practice setting, but I am unclear what the specialties were. 

 a/ The reason this is important is that other data have suggested that specialty area of practice has influenced the uptake of TH. 

2/ The methods section is entirely too long and detracts from the findings of the manuscript. The investigative team should strongly consider revisions to this section. This is not a methods driven manuscript. 

3/ The results section is a bit hard to get through with the embedded tables. But these tables are the actual important findings that should have a bit more text added to the results section. 

4/ Given the uncertainty of the participant specialties, the conclusions may be a bit overly broad.

Reviewer #2: PDIG-D-24-00296

Barriers and facilitators to physicians’ telemedicine uptake during the beginning of the

COVID-19 pandemic

This article surveyed a range of physicians in the US to determine their views on the use of telemedicine at the beginning of the COVID Pandemic. This is an important and relevant question as the role of telemedical consults has grown recently. Their questions were based on literature review and are appropriate. The sampling process to recruit subjects for the survey had limitations, although using a broader survey on mental health care to de-emphasize the telemedicine aspect could have strengthened the recruitment of a broader range of practitioners not just those interested in technology. Overall this paper includes valuable insights but there are a number of issues that need to be addressed. 

Abstract:

Good, on first reading. Clear relevant points, not structured. 

More information is needed on sampling methods, and lack of racial and ethnic representation. 

Authors should include the statistical analyses used and the significant results found.

Introduction:

Good clear statement of research questions. 

Survey was sent to 850 by email, 46 were undelivered, 315 opened survey, 228 with full data were included. 

There was a low percentage of black and Latino physicians and those from rural areas.

Details of survey participants should be in the results section.

Survey occurred at the beginning of the COVID pandemic so likely reflected experiences before COVID and initial scale up with existing technologies. 

Results:

Table 1: 

Needs N in table

Percentages only need 1 decimal place

Percentage calculations are wrong, for example in section on Race/ethnicity and section on Primary Practice Setting

Full table needs to be recalculated and redesigned for clarity.

Survey questions appear appropriate with free text options

Table 2:

Structure needs to be improved to link titles and responses clearly (table lines?)

Response rate of 62/228 (27%) is relatively low. 

The table is large and gives no indication of the frequency of different coded responses. 

It would be clearer to summarize the most common types of free text responses in the results and include a full table of all responses and numbers in each category in an appendix. 

Tables 4 and 5:

These tables are largely redundant as few of the odds ratios are significant at the P< .05 level. They should be summarized in the text or in much smaller tables that just highlight the significant values and perhaps indicates those that are of borderline significance. The full tables could be in appendices. It would be helpful to indicate to the reader that Exp(B) is the odds ratio. 

Table 6:

This table only shows 2 statistically significant relationships and could be simplified with clear highlighting of the significant relationships, or moved to an appendix with significant results in the text. 

Discussion:

Clear and helpful with important points about limitations to the survey. 

The sampling process to recruit participants clearly had limitations in terms of responses. Do the authors have any evidence as to whether this may have favored certain groups?

Do the authors believe the results would be different 1-2 years after the start of the pandemic when the technology, training, experience and legal and economic structures had changed?

Limitations of the statistical analyses also need to be discussed particularly the relatively small sample size for the logistic regression analysis and the small number of statistically significant relationships.

Reviewer #3: Thank you for submitting your manuscript on barriers and facilitators to physicians' telemedicine uptake during the COVID-19 pandemic. Your study provides valuable insights into this very intense topic. 

The mixed-methods approach and substantial sample size are strengths. However, there are several areas that require attention before publication. The structure could be improved by moving the explanation of survey options earlier in the methods section. The results presentation, particularly Table 4, is overwhelming and could benefit from a more digestible format, such as smaller tables or visual representations. While the discussion is comprehensive, it could be strengthened by more explicit connections to prior literature and deeper exploration of policy implications. The limitations section is candid but could further address how these limitations might influence result interpretation, especially regarding the COVID-19 context's impact on generalizability. Additionally, the low Nagelkerke pseudo-R squared values warrant more explicit discussion. Addressing these points will enhance the manuscript's clarity, rigor, and impact. We look forward to seeing a revised version that builds on the strong foundation you've established.

6. PLOS authors have the option to publish the peer review history of their article (what does this mean?). If published, this will include your full peer review and any attached files.

**Do you want your identity to be public for this peer review?** For information about this choice, including consent withdrawal, please see our Privacy Policy.

Reviewer #1: No

Reviewer #2: No

Reviewer #3: No

---

## [Decision Letter · Decision Letter 1]

13 Dec 2024

PDIG-D-24-00296R1Barriers and facilitators to physicians’ telemedicine uptake during the beginning of the COVID-19 pandemicPLOS Digital Health Dear Dr. Perrin, Thank you for submitting your manuscript to PLOS Digital Health. After careful consideration, we feel that it has merit but does not fully meet PLOS Digital Health's publication criteria as it currently stands. Therefore, we invite you to submit a revised version of the manuscript that addresses the points raised during the review process. Please submit your revised manuscript within 60 days Feb 11 2025 11:59PM. If you will need more time than this to complete your revisions, please reply to this message or contact the journal office at digitalhealth@plos.org. Please include the following items when submitting your revised manuscript:* A rebuttal letter that responds to each point raised by the editor and reviewer(s). You should upload this letter as a separate file labeled 'Response to Reviewers'. This file does not need to include responses to any formatting updates and technical items listed in the 'Journal Requirements' section below.* A marked-up copy of your manuscript that highlights changes made to the original version. You should upload this as a separate file labeled 'Revised Manuscript with Track Changes'.* An unmarked version of your revised paper without tracked changes. You should upload this as a separate file labeled 'Manuscript'. If you would like to make changes to your financial disclosure, competing interests statement, or data availability statement, please make these updates within the submission form at the time of resubmission. Guidelines for resubmitting your figure files are available below the reviewer comments at the end of this letter. We look forward to receiving your revised manuscript. Kind regards, Calvin Or, PhDSection EditorPLOS Digital Health Calvin OrSection EditorPLOS Digital Health Leo Anthony CeliEditor-in-ChiefPLOS Digital Healthorcid.org/0000-0001-6712-6626 **Additional Editor Comments (if provided):****Reviewers' Comments:** Reviewer's Responses to Questions

**Comments to the Author**

1. If the authors have adequately addressed your comments raised in a previous round of review and you feel that this manuscript is now acceptable for publication, you may indicate that here to bypass the “Comments to the Author” section, enter your conflict of interest statement in the “Confidential to Editor” section, and submit your "Accept" recommendation.

Reviewer #4: All comments have been addressed

Reviewer #5: (No Response)

Reviewer #6: All comments have been addressed

Reviewer #7: (No Response)

Reviewer #8: (No Response)

Reviewer #9: (No Response)

2. Does this manuscript meet PLOS Digital Health’s publication criteria? Is the manuscript technically sound, and do the data support the conclusions? The manuscript must describe methodologically and ethically rigorous research with conclusions that are appropriately drawn based on the data presented.

Reviewer #4: Yes

Reviewer #5: Partly

Reviewer #6: Yes

Reviewer #7: Partly

Reviewer #8: Yes

Reviewer #9: Partly

3. Has the statistical analysis been performed appropriately and rigorously?

Reviewer #4: Yes

Reviewer #5: Yes

Reviewer #6: Yes

Reviewer #7: Yes

Reviewer #8: Yes

Reviewer #9: Yes

4. Have the authors made all data underlying the findings in their manuscript fully available (please refer to the Data Availability Statement at the start of the manuscript PDF file)?

Reviewer #4: No

Reviewer #5: (No Response)

Reviewer #6: Yes

Reviewer #7: Yes

Reviewer #8: Yes

Reviewer #9: Yes

5. Is the manuscript presented in an intelligible fashion and written in standard English?

Reviewer #4: Yes

Reviewer #5: (No Response)

Reviewer #6: Yes

Reviewer #7: Yes

Reviewer #8: Yes

Reviewer #9: Yes

6. Review Comments to the Author

Reviewer #4: Please add a statement about data availability. As mentioned in this journal, PLOS Data policy requires authors to make all data underlying the findings described in their manuscript fully available without restriction, with rare exception. The data should be provided as part of the manuscript or its supporting information, or deposited to a public repository.

Reviewer #5: This is an interesting manuscript that explores potential clinician-perceived barriers and facilitators or telehealth at the start of the pandemic. Although somewhat dated, it is still valuable to know what demographics may be particularly likely or unlikely to adopt telehealth to inform education efforts.

Previous reviewer comments on the unclear structure of the results and the inclusion of excessive detail have been satisfactorily addressed. However, some concerns remain to do with the relevance of findings in the 2025 context (and how well the discussion addresses this) and limitations of the qualitative approach.

The paragraph beginning with “Identifying the most and least endorsed barriers and facilitators can help inform institutional policies” requires more nuanced consideration given much has changed since the time of the survey. This should be a larger focus of the discussion, not just relegated to the limitations section. I’m aware that you also commented about the consistency of the findings during the pandemic, but post-pandemic there have been many changes in reimbursement, access to improved technology, improving digital literacy, literature on the lack of impact of digital delivery on quality of care or therapeutic alliance. Your discussion could be strengthened by consideration of which factors are likely to persist in the current environment as that will be more informative for the design of interventions.

Qualitative methodologies are broad and you have not specified what approach was used (thematic? Content?) or the coding approach taken (semantic? Inductive? Deductive?)? Which staff were raters? Without adequate description of the qualitative methods it is hard to judge quality and alignment with existing approaches.

It seems potentially problematic to simply recode the free text responses under the existing options - presumably, the respondents felt like these didn’t adequately capture their perspectives, hence they provided an “other” response. And this lacks some of the richness that would be expected of a qualitative report. I found the absence of example comments disappointing as they could provide useful elaboration, for example, on why the quality of care or the relationship was impacted. I’m not sure why the “miscellaneous” category was included, what does this add to the analysis when there is no coherent idea linking these responses?

A potential limitation is integrating the free text responses into the quantitative analysis. If a physicians didn’t mention, for example, inadequate patient technological literacy in a free text comment, it doesn’t mean they do not perceive this to be important. It may have simply meant they were not motivated enough or too time poor to comment. They may have selected these if they were presented in the survey. This may give the reader an inaccurate impression that this barrier was not perceived as important - rather, it was not asked after. This should be acknowledged as a limitation especially in light of literature showing the impacts of the digital divide on access to healthcare.

Reviewer #6: Great work on addressing the comments and improving the readability of the paper!

Minor thoughts/questions:

* Participants (p. 7): What was the geographical focus of this study or what physicians in the USA did you invite to participate? - might be worth mentioning this first thing to not cause any confusion later on (since the N would be quite small for all US physicians).

* Procedures (p. 7): Since that's a short sub-section and somewhat overlaps with previous information, I wondered if it would be useful to merge this section with "participants"?

* Data Analysis (p. 8): What does "endorsing" barriers/facilitators mean? Merely selecting them in the survey (and those items that were not selected were automatically "not endorsed")?

* Results (p. 12): are the statistical measures you present a p-value for ORs? (e.g.: "Older physicians were less likely to endorse inefficient use of time (p < .001) and potential for medical errors (p = 0.034) as barriers to telemedicine use compared to younger physicians." - it might be worth mentioning the measurement as they are part of the supplementary material but not the "main paper".

* Discussion (p. 18): Personal request: Would you mind mentioning "allied health professionals" instead of singling out PTs in the last sentence of the discussion?

Reviewer #7: Thank you for the opportunity to review this manuscript describing barriers and facilitators for the use of telemedicine by physicians at the beginning of the COVID-19 pandemic. The authors have tackled an important topic, provided a strong rationale for the investigation, and thoughtful discussion of the implication of results. They have improved this manuscript through revisions and suggestions from reviewers and it appears easier to read and understand findings. Although I do believe this manuscript overall would add to the body of literature for this topic, I believe that there are several areas that could be revised to enhance clarity and ensure the manuscript accurately reflects the findings, given the statistical analyses used. Thus, my comments are largely focused on the results section and ensuring that significant findings are appropriately explained in the context of all analyses conducted. I believe these revisions will strengthen the manuscript and allow readers to better interpret these findings. Below, I have provided detailed feedback organized by section for the authors.

Introduction/Discussion

- The introduction may benefit from slight restructure to include the importance of telemedicine and factors associated with sustained use (lines 97 through 103) prior to review of barriers and facilitators of use (lines 88 through 96). This would help improve and establish the rationale for examining these factors and their importance (even as we move into “post” COVID-19 healthcare).

Methods

- Do you have more information regarding the sampling procedures used? I wonder what percentage of the 850 recruitment announcements sent were individuals identified through hospital and health clinic websites and were these hospitals national or local to the researcher’s institution? How were these hospitals and health clinics selected? This may help explain and understand the physician characteristics and help understand generalizability to other physicians in the United States.

Results

- What does it mean if the omnibus binary logistic regressions were not significant? Does that impact interpretation of the predictors?

- Given these are multivariable logistic regressions, statements of results and significance of variables (e.g., age) should be stated “within the context of all other variables included in the model”.

- There were 16 separate logistic regressions run in this study. 4 out 8 barrier analyses yielded no significant results and 7 of 8 facilitator analyses yielded no significant results. Please add explanation to the results section detailing null findings and comment in the discussion as to why these predictors may not have explained variation in endorsement of a particular barrier/facilitator.

Discussion

- Be careful with language on line 236. It seems like identifying barriers and facilitators could inform institutional policies aimed at increasing adoption or sustainment of use of telemedicine… but I’m not sure about funding.

- The interpretation of the Nagelkerke pseudo-R2 statistics (lines 251 through 253) is misleading, as only two models were significant (and meaningfulness is up for interpretation or defense). This should be revised to be more accurate.

Tables

- Table S1 is still confusing, particularly the note. The table displays all codes used in the study, but the note makes it appear that these were all generated based on qualitative responses. Consider providing a general description of table and then specific notes including more information for clarity (e.g., “Descriptions of barriers and facilitators created based on research team agreement. Physicians’ “other” responses resulted in 20 facilitators and 43 barriers that were coded into the following categories: lack of patient access (n = 9), …”).

- Denote on Table S1 the three categories added to the list of researcher-generated codes.

- Consider landscape format for Table S1, consistency in text alignment within cells, and adding spacing to increase readability.

Reviewer #8: This paper provides useful insight into the barriers and facilitators, and contextual characteristics influencing these factors, associated with telemedicine use in the US. The authors have addressed reviewer comments well however, some revisions to reviewer comments could be addressed better and there are a couple of minor additional points that should be considered.

1) Addressing previous reviewer comments:

a) while the methods section has been slightly shortened, the description of the qualitative analysis could be further shortened/clarified to explain what the raters did i.e. what qualitative analysis strategy did the authors use? Looking at the reference it seems to be a qualitative content analysis. This could be presented much more simply e.g. the raters conducted independent content analysis to identify and categorise additional barriers/facilitators. Also the results of this qualitative analysis should be presented in the results rather than the methods.

b) the authors have moved the main results tables to supplementary materials per previous reviewers comments that the tables were too long, however now there are no main results tables presented in the results section. It would be useful to have a shorter table to substantiate claims on some of the main/significant findings (e.g. S2), and then the supplementary materials could be referenced for full results.

2) Additional points relating to methods (data analysis section):

a) the grouping of practice characteristics requires further explanation/justification. For medical centers vs other settings, please clarify what the definition of medical centers was in the current study e.g. was a hospital considered a medical center? For urban vs suburban and rural settings, please either justify why suburban and rural were grouped or perhaps consider referencing the grouping of these settings as a limitation. Typically, urban and suburban would be grouped as rural settings possess unique characteristics. Also, the results could highlight that there were no stat significant results for urban vs suburban and rural (and limitations could highlight that this could be due to the urban/suburban not being much different contextually).

b) Additionally, the data analysis section does not describe why chi squared tests and Nagelkerke pseudo-R squared were run, some brief additional detail on this would be useful.

Reviewer #9: This study examines telemedicine adoption during the COVID-19 pandemic, a critical period of healthcare transformation. The analysis of key barriers and facilitators offers practical insights for healthcare policy and telemedicine implementation strategies. However, the manuscript requires significant revisions before it can be considered for publication:

Logistic Regression Models

- The paper does not adequately explain why specific variables were chosen for the logistic regression models, weakening the rationale behind the analysis.

- The logistic regression models have low explanatory power (e.g., low Nagelkerke pseudo-R² values), indicating that critical factors influencing barriers and facilitators might be missing.

- Important variables such as years in practice, the number of providers in a practice, and additional practice settings were not included, despite their potential relevance to telemedicine adoption.

- The authors should incorporate these variables and explicitly justify their inclusion or exclusion.

Discussion Section

- The findings are not sufficiently grounded in prior literature, reducing the strength of the discussion.

- Specific Gaps in Citations:

Lines 221-227: No references are provided to support differences in telemedicine adoption between older and younger physicians.

Lines 232-236: The claim that physicians may be frustrated by the lack of in-person resources or that telemedicine reduces patient costs due to avoided ancillary fees lacks supporting citations.

Lines 240-248: While the authors appropriately cite literature on female providers spending more time with patients, they do not provide evidence for gender-based differences in telemedicine appointment durations or implementation challenges.

Minor Feedback

- Remove categories/levels with N=0, %=0 in Table 1 (e.g., Correctional Facility, Geriatric Facility).

- Separate Tables 4 and 5 into individual tables with titles reflecting outcomes (e.g., "Better Access to Care").

- Move chi-squared and Nagelkerke pseudo-R² values from Table 6 into corresponding tables for logistic regression results.

By addressing these issues, the manuscript will better justify its methodology, improve the integration of findings into existing literature, and enhance its clarity and impact.

7. PLOS authors have the option to publish the peer review history of their article (what does this mean?). If published, this will include your full peer review and any attached files.

**Do you want your identity to be public for this peer review?** For information about this choice, including consent withdrawal, please see our Privacy Policy.

Reviewer #4: **Yes: **John Robert Bautista

Reviewer #5: No

Reviewer #6: No

Reviewer #7: No

Reviewer #8: No

Reviewer #9: No

---

## [Decision Letter · Decision Letter 2]

8 Mar 2025

Barriers and facilitators to physicians’ telemedicine uptake during the beginning of the COVID-19 pandemic

PDIG-D-24-00296R2

Dear Dr. Perrin,

We are pleased to inform you that your manuscript 'Barriers and facilitators to physicians’ telemedicine uptake during the beginning of the COVID-19 pandemic' has been provisionally accepted for publication in PLOS Digital Health.

Best regards,

Calvin Or, PhD

Section Editor

PLOS Digital Health

**Additional Editor Comments (if provided):**

**Reviewer Comments (if any, and for reference):**

Reviewer's Responses to Questions

**Comments to the Author**

1. If the authors have adequately addressed your comments raised in a previous round of review and you feel that this manuscript is now acceptable for publication, you may indicate that here to bypass the “Comments to the Author” section, enter your conflict of interest statement in the “Confidential to Editor” section, and submit your "Accept" recommendation.

Reviewer #5: All comments have been addressed

Reviewer #6: All comments have been addressed

Reviewer #8: All comments have been addressed

2. Does this manuscript meet PLOS Digital Health’s publication criteria? Is the manuscript technically sound, and do the data support the conclusions? The manuscript must describe methodologically and ethically rigorous research with conclusions that are appropriately drawn based on the data presented.

Reviewer #5: Yes

Reviewer #6: Yes

Reviewer #8: Yes

3. Has the statistical analysis been performed appropriately and rigorously?

Reviewer #5: Yes

Reviewer #6: Yes

Reviewer #8: Yes

4. Have the authors made all data underlying the findings in their manuscript fully available (please refer to the Data Availability Statement at the start of the manuscript PDF file)?

Reviewer #5: Yes

Reviewer #6: Yes

Reviewer #8: Yes

5. Is the manuscript presented in an intelligible fashion and written in standard English?

Reviewer #5: Yes

Reviewer #6: Yes

Reviewer #8: Yes

6. Review Comments to the Author

Reviewer #5: I'm satisfied that the authors have addressed the issues I and other authors previously commented on, by providing the required detail, or acknowledging the issue in the limitations.

Reviewer #6: (No Response)

Reviewer #8: Thank you for your responses and edits to the manuscript, reviewer comments have been thoroughly addressed and I have no further comments.

7. PLOS authors have the option to publish the peer review history of their article (what does this mean?). If published, this will include your full peer review and any attached files.

**Do you want your identity to be public for this peer review?** For information about this choice, including consent withdrawal, please see our Privacy Policy.

Reviewer #5: No

Reviewer #6: No

Reviewer #8: No
